

# Could resistance to lactate accumulation contribute to the better swimming performance of *Brycon amazonicus* when compared to *Colossoma macropomum*?

Marcio S. Ferreira[1], Paulo H.R. Aride[2] and Adalberto L. Val[1]

[1] Laboratory of Ecophysiology and Molecular Evolution, Brazilian National Institute for Research of the Amazon, Manaus, Amazonas, Brazil
[2] Laboratory of Nutrition and Aquatic Organisms Production, Federal Institute of the Espírito Santo State, Piúma, Espírito Santo, Brazil

## ABSTRACT

**Background**. In the wild, matrinchã (*Brycon amazonicus*) and tambaqui (*Colossoma macropomum*) rely strongly on their swimming capacity to perform feeding, migration and reproductive activities. Sustained swimming speed in fishes is performed almost exclusively by aerobic red muscles. The white muscle has high contraction power, but fatigue quickly, being used mainly in sprints and bursts, with a maximum duration of few seconds. The Ucrit test, an incremental velocity procedure, is mainly a measure of the aerobic capacity of a fish, but with a high participation of anaerobic metabolism close to the velocity of fatigue. Our previous study has indicated a high swimming performance of matrinchã (Ucrit) after hypoxia exposure, despite increased levels of lactate in plasma. In contrast, tambaqui with high lactate levels in plasma presented very low swimming performance. Therefore, we aimed to study the resistance of matrinchã and tambaqui to the increased lactate levels in muscle over an incremental velocity test (Ucrit). As a secondary aim, we analyzed the differences in anaerobic metabolism in response to environmental hypoxia, which could also support the better swimming performance of matrinchã, compared to tambaqui.

**Methods**. We measured, over incremented velocities in both species, the metabolic rate (the oxygen consumption by the fish; $MO_2$), and the concentrations of lactate and nitrites and nitrates (NOx) in muscles. NOx was measured as an indicator of nitric oxide and its possible role in improving cardiorespiratory capacity in these fishes, which could postpone the use of anaerobic metabolism and lactate production during the swimming test. Also, we submitted fishes until fatigue and hypoxia (0.5 mg $L^{-1}$) and measured, in addition to the previous parameters, lactate dehydrogenase activity (LDH; the enzyme responsible for lactate production), since that swimming performance could also be explained by the anaerobic capacity of producing ATP.

**Results**. Matrinchã exhibited a better swimming performance and higher oxygen consumption rates. Lactate levels were higher in matrinchã only at the moment of fatigue. Under hypoxia, LDH activity increased in the white muscle only in tambaqui, but averages were always higher in matrinchã.

**Discussion and conclusions**. The results suggest that matrinchã is more resistant than tambaqui regarding lactate accumulation in muscle at the Ucrit test, but it is not clear how much it contributes to postpone fatigue. The higher metabolic rate possibly

Corresponding author
Marcio S. Ferreira,
marcio.ferreira@inpa.gov.br,
marcioferreira.inpa@gmail.com

allows the accumulated lactate to be used as aerobic fuel by the matrinchã, improving swimming performance. More studies are needed regarding matrinchã's ability to oxidize lactate, the effects of exercise on muscle acidification, and the hydrodynamics of these species, to clarify why matrinchã is a better swimmer than tambaqui.

## INTRODUCTION

Most fishes lack defenses against predators; thus, swimming is the primary method for avoiding and surviving attacks (*Watkins, 1996*). Furthermore, it is assumed that the maximal swimming performance may strongly influence the ability of a fish to obtain food, find a mate, and avoid other unfavorable conditions (*Drucker, 1996*). Sustained swimming speed is performed without resulting in muscular fatigue by the almost exclusive use of aerobic red muscles (*Webb, 1997*). It includes the cruising speed of migrating fishes and speeds for routine activity, like spontaneous swimming, foraging, and position holding. The white muscle has high contraction power, but fatigue quickly, being used mainly in sprints and bursts (e.g., to catch prey or to avoid a predator attack), with a maximum duration of few seconds. Red muscle can be used in conjunction with white muscle, but their contribution to power generation is low (*Reidy, Kerr & Nelson, 2000*).

The Amazonian fish species tambaqui (*Colossoma macropomum*) and matrinchã (*Brycon amazonicus*) are important species for aquaculture (*Oliveira et al., 2012*). Despite differences in body shape, matrinchã is a fusiform and tambaqui is a rounded fish, in the wild, they rely strongly on their swimming capacity to perform feeding, migration and reproductive activities (*Goulding, 1981*; *Goulding & Carvalho, 1982*; *Santos, Ferreira & Zuanon, 1998*). Even on the distribution of red and white muscle along with the longitudinal axis, they look like similar (personal observation).

However, our previous study (*Ferreira, Oliveira & Val, 2010*) have indicated a high swimming performance of matrinchã Ucrit; (*Brett, 1964*) compared to tambaqui. The Ucrit test (*Brett, 1964*), an incremental velocity procedure, is mainly a measure of aerobic capacity of a fish, but with a high participation of anaerobic metabolism close to velocity of fatigue (*Hammer, 1995*; *Burgetz et al., 1998*; *Richards et al., 2002*; *Peake & Farrell, 2004*). Also, in that work, we saw that tambaqui presented a higher Ucrit even with high levels of lactate in the plasma resulting from environmental hypoxia. Therefore, based on those results, we suggested that matrinchã might have low sensitivity to lactate accumulation in muscle, and this could explain its capacity of keep swimming under these circumstances. However, muscle lactate was not measured, and this suggestion was made speculatively.

In contrast, another of our previous study (*Ferreira, 2006*), tambaqui was shown to not swim in the Ucrit test; (*Brett, 1964*) if previously exposed to environmental hypoxia, also with high lactate levels in plasma. Unfortunately, we have no measured muscle lactate in

that experiment too, making it difficult to try explaining such differences in swimming performance between tambaqui and matrinchã after hypoxia.

The literature points out that when hypoxia levels are severe and persistent, tambaqui decreases its aerobic metabolism and improves anaerobic capacity as a strategy for surviving for a long time under these circumstances, much common in the Amazon waters (*Val, 1995*; *Val, 1996*; *Val, Silva & Almeida-Val, 1998*). However, a depressed and predominantly anaerobic metabolism could impose a severe restriction on the energetic budget and sounds reasonable that this could explain the reduction in swimming capacity of tambaqui after hypoxia.

*Gladden (2004)*, reviewing the effects of lactate in animals in general, pointed that increased levels of muscle $[H^+]$ could depress its function by "(1) reducing the transition of the cross-bridge from the low- to the high-force state, (2) inhibiting maximal shortening velocity, (3) inhibiting myofibrillar ATPase, (4) inhibiting glycolytic rate, (5) reducing cross bridge activation by competitively inhibiting $Ca^{2+}$ binding to troponin C, and (6) reducing $Ca^{2+}$ re-uptake by inhibiting the sarcoplasmic ATPase (leading to subsequent reduction of $Ca^{2+}$ release)".

As reviewed by *Webb (1997)*, the swimming performance of fish is directly related to its ability to transfer oxygen and energy molecules to the muscles, and at the same time, eliminating undesirable by-products. At least in humans, it is believed that nitric oxide generated in exercised muscles helps to delay fatigue, since it vasodilates and improves delivery of oxygen and nutrients to the muscles and removes $CO_2$ (*Macardle, 2011*). Nitric oxide is produced from L-arginine by the enzyme nitric oxide synthase (NOS) in a process that requires NADPH and $O_2$ (*Alderton, Cooper & Knowles, 2001*). Under intense exercise, the physiological stimulus to increase production of endothelial nitric oxide is the friction caused by increased blood flow through the vessel lumen (*Pohl et al., 1986*; *Rubanyi, Romero & Vanhoutte, 1986*; *Hutcheson & Griffith, 1991*). Due to its evanescence, the detection of nitric oxide is tricky and usually made indirectly, considering the measurement of the concentration of nitrates and nitrites (NOx) through a relatively simple technique (*McNeill & Perry, 2006*).

Based on our previous studies that matrinchã can keep swimming even with high lactate levels in the plasma, in the present work our focus was to verify if and why matrinchã is more resistant than tambaqui regarding lactate accumulation caused by graded exercise until fatigue (Ucrit test). Also, we focused on understanding the differences in anaerobic metabolism between both species exposed to environmental hypoxia. Such data would help us to identify the traits that support the better swimming performance of matrinchã, compared to tambaqui.

The first hypothesis is that due to increases in exercise intensity, tambaqui does not accumulate lactate faster than matrinchã and, therefore, fatigue in these species is not related to differences in time taken to attain high and equivalent levels of lactate in muscles. The second hypothesis is that, at the moment of fatigue, both species have similar lactate levels in muscle, indicating that, contrary to what we expected, resistance to lactate accumulation does not contribute for the better swimming performance of matrinchã. The third hypothesis is that the muscle anaerobic metabolism is similar between these

species under environmental hypoxia or at fatigue, indicating that their capacity of dealing with extreme conditions of muscle anaerobiosis does not contribute for the differences in swimming performance.

To test these hypotheses, we measured, over incremented velocities in both species, the concentrations of lactate, the metabolic rate (the oxygen consumption by the fish; $MO_2$), and the NOx (as an indicator of nitric oxide) in muscles. NOx and $MO_2$ were measured to help us to understand how the limitations or improvements of aerobic capacity are related to the use of anaerobic metabolism and lactate accumulation in tambaqui and matrinchã. Also, we submitted fishes until fatigue and hypoxia and measured, in addition to the previous parameters, lactate dehydrogenase activity (the enzyme responsible for lactate production), since that possible differences in swimming performance between these two species could also be supported by their anaerobic capacity of producing ATP.

## MATERIAL & METHODS

### Experimental fishes and water parameters

Specimens of matrinchã [*Brycon amazonicus* (Spix and Agassiz, 1829) and tambaqui (*Colossoma macropomum* (Cuvier 1818)), respectively weighing $15.20 \pm 2.45$ g and $31.00 \pm 6.99$ g and measuring $8.80 \pm 0.82$ cm and $9.87 \pm 0.71$ cm, were purchased from local fish farmers (Sítio dos Rodrigues, Km 35, Rod. AM-010, Brazil) and kept in continuously aerated water, renewed daily by 70%, prior to the experiments. The physicochemical characteristics of well water used in the experiments were: $Na^+$ $34.0 \pm 1.0$ μmol, $Cl^-$ $28.0 \pm 1.0$ μmol, $Ca^{2+}$ $11.5 \pm 0.9$ μmol, $Mg^{2+}$ $0.80 \pm 0.10$ μmol, $K^+$ $15.5 \pm 0.4$ μmol, pH $6.32 \pm 0.04$, dissolved $O^2$ $5.91 \pm 0.07$ mgL$^{-1}$, temperature $28.0 \pm 1.0$ °C. Fish were fed a commercial diet (Nutripiscis-Presence® AL 45, SP Rações, São Paulo, SP, Brazil) containing 36% gross protein, up to apparent satiation, twice a day. Gaseous ammonia in the water (estimated from measurements of pH and total ammonia) was monitored and did not increase above 0.02 mg L$^{-1}$. Experimental work was approved by the Ethics Committee on Animal Experiments of INPA (protocol number: 020/2013) and conformed to national animal care regulations.

### Metabolic rate, lactate and NOx production at submaximal velocities and fatigue

The fatigue test was performed based on the Ucrit test of *Brett (1964)*. In addition to the Ucrit test, which the same fish was exposed to all velocities until fatigue, we made other trials, using different fishes in each velocity so that we could collect tissues of the fishes in each effort step. Six fish ($n = 6$) for each velocity or fatigue were individually analyzed in closed 5-L swimming chambers (model #SW10060; Loligo Systems, Tjele, Denmark), with a built-in sensor for monitoring oxygen and temperature levels. After one hour of acclimation, water speed was increased by 1.25 body length per second, until the fish swam for 30 min at the target velocity. At each velocity, two cycles of oxygen measurements, followed by two cycles of flushing with oxygenated water, were completed. Slope means for the two measurements were used for the calculation of metabolic rate [rate of oxygen consumption $(MO_2)$], as recommended by Loligo Systems®. For the subsequent velocity, new fish swam

all previous velocities until reaching the target speed, and was sacrificed to analyze white muscle for lactate and red muscle for NOx. The recovery was tested using new fish, which swam at 0.1 bl s$^{-1}$ for 60 min, after passing through all previous submaximal velocities.

A fish was considered fatigued when it could no longer maintain its position in the swimming chamber, even after one electrical stimulation (5 V). At the time of fatigue, critical swimming speed was determined for each fish, using the equation described by *Brett (1964)*:

$$\text{Ucrit} = V_{-1} + (t/\Delta t)\Delta v$$

where $\Delta t$ is the time increment (min), $\Delta v$ is the velocity increment (bl s$^{-1}$), $t$ is the time elapsed at the final velocity (min), and $V_{-1}$ is the highest velocity maintained for the prescribed period (bl s$^{-1}$). Results are reported as body lengths per second (bl s$^{-1}$).

After each effort step or after fatigue, fishes were anesthetized by immersion in water containing 0.1 g L$^{-1}$ of MS-222 (tricaine methanesulfonate, Sigma Chemical, pH corrected with NaOH), and sacrificed by a cervical section of the spinal cord. Samples of red and white muscle, near the caudal peduncle, were then collected with a surgical blade and frozen at $-80$ °C. Tissues (both muscles types) were removed for analyses of maximum activity of the enzyme lactate dehydrogenase (LDH) only in resting and fatigued fishes. For this measurement, we used the method of *Driedzic & Almeida-Val (1996)*, after tissue homogenization in imidazole cold buffer at 150 mM (1 mM EDTA, 5 mM dithiothreitol and 1% triton X-100, pH 7.4). Following homogenization, samples were centrifuged at $13,000\times$ g for 15 min (Eppendorf centrifuge, model R404A), and the supernatant was collected for analyses.

For all fishes, we have determined the lactate concentration in white muscle, according to *Gutmann & Wahlefeld (1974)*, after tissue homogenization in perchloric acid at 8% and collection of the supernatant, after centrifugation as described for LDH. For the estimation of nitric oxide in red muscle, we made an indirect measurement, since that, due to its evanescence, the detection is tricky. For that, we measured the concentration of nitrates and nitrites (NOx) (*McNeill & Perry, 2006*) in red muscle using a commercial kit. The red muscle was homogenized in phosphate buffer (10 mM PO$_4^{3-}$, 137 mM NaCl, and 2.7 mM KCl) and centrifuged in filters (Millipore©Amicon 45 kDa) for sample deproteinization, prior determinations as recommended by the manufacturer (Item 780001; Cayman Chemical, Ann Arbor, MI, USA).

## Hypoxia exposure test

Twelve fish of each species were individually kept in aerated 2-L experimental chambers for 24 h for acclimation. After 12 h, 70% of the water was replaced. At this point, the aeration of six chambers of each species was turned off, and nitrogen gas was injected into the water until the oxygen concentration was lowered to 0.5 mg L$^{-1}$. The remaining fish (six of each species) were maintained in normoxia (6 mg L$^{-1}$). In the experimental chambers, the target hypoxia level was reached in 30 min. A plastic sheet placed on the water surface prevented aquatic surface respiration (ASR). Fish were kept in this condition for one hour and then were euthanized. Red and white muscles of all fishes were harvested and analyzed for LDH, NOx, and lactate, as described above.

## Statistical analysis

Data are expressed as mean $\pm$ sem ($n = 6$). Statistical analysis of the data was conducted using the SigmaStat 3.5 software (Systat Software Inc., San Jose, CA, USA). For LDH in the swimming experiment (Ucrit test) and all parameter in the hypoxia experiment, the statistical significance of differences between the means was estimated by two-factor analysis of variance (ANOVA) for muscle type and species. For oxygen consumption, lactate and NOx in the swimming experiment (Ucrit test), we applied one-factor ANOVA for each species. The Student-Newman-Keuls (SNK) method was used as a posthoc test, and a significance level of 5% was employed throughout. All data were tested regarding ANOVA assumptions.

## RESULTS

### Lactate, metabolic rate and NOx production at submaximal velocities and fatigue

Matrinchã exhibited a higher ($p < 0.05$) swimming performance compared to tambaqui. The measured Ucrit values were $6.10 \pm 0.80$ bl s$^{-1}$ and $3.50 \pm 0.35$ bl s$^{-1}$ for matrinchã and tambaqui, respectively.

Lactate values in white muscle of tambaqui were not higher than resting fishes any time but are higher ($p < 0.05$) than values at the first velocity (1.25 bl s$^{-1}$) at 2.5 bl s$^{-1}$ (Fig. 1). In matrinchã, lactate values were not higher than control fishes until fatigue, but if we compare to the first velocity (1.25 bl s$^{-1}$), the lactate levels are different starting from 3.75 bl s$^{-1}$. The lactate levels felt down to control levels after 1 h of recovery at rest, for both species.

The average rate of oxygen consumption was higher in matrinchã compared to tambaqui over the entire experimental period, even before exercise (Fig. 2). The maximum rate of oxygen consumption was 13.92 $\mu$mol g$^{-1}$h$^{-1}$ in tambaqui at 2.5 bl s$^{-1}$ (body lengths per second), and 18.72 $\mu$mol g$^{-1}$ h$^{-1}$ in matrinchã at 5 bl s$^{-1}$. For both species, these were the sustained swimming velocities closest to that of fatigue, in which all animals were able to complete.

The velocities in which the fishes fatigued were not considered in MO$_2$ calculations since that is necessary the fish swim the entire period (30 min) for oxygen consumption rate be quantified. In tambaqui, oxygen consumption at 2.5 bl s$^{-1}$ was higher ($p < 0.05$) than at resting levels (0.1 bl s$^{-1}$), fatiguing soon after. In matrinchã, the MO$_2$ was higher ($p < 0.05$) than control levels at 3.75, increasing even more at 5.0 bl s$^{-1}$, just before the fatigue time. Oxygen consumption returned to the levels that were not significantly different from resting rates within 15 min in recovered animals of both species (Fig. 2).

NOx levels in red muscle were increased ($p < 0.05$; Fig. 3) in fatigued fish of both species compared to resting fish, and NOx levels were higher in matrinchã compared to tambaqui, at the moment of fatigue. Fatigued tambaqui presented levels of NOx in red muscles 80% higher than in the control animals ($p < 0.05$), while the increase in fatigued matrinchã was 96% higher ($p < 0.05$). In tambaqui, at the velocity of 2.5 bl s$^{-1}$, the NOx levels are already higher than control fishes ($p < 0.05$), keeping this trend at the moment of fatigue.

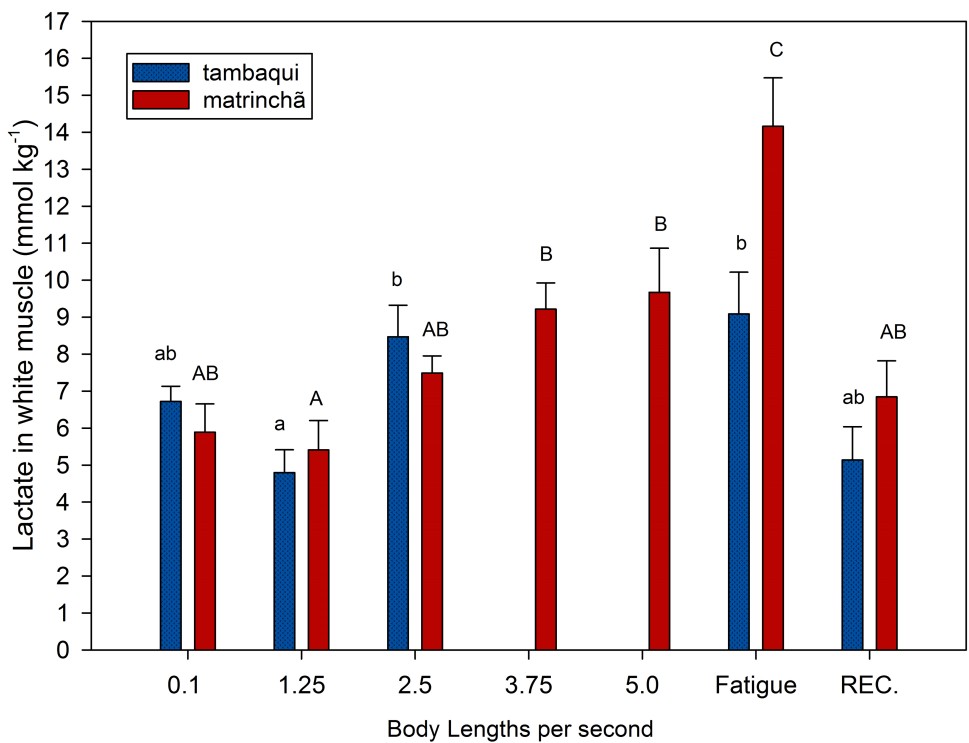

**Figure 1 Lactate in white muscle of tambaqui and matrinchã exposed for 30 minutes to submaximal velocities (body lengths per second; bl s$^{-1}$) and at fatigue.** Recovery (REC.) refers to the new fish swimming at 0.1 bl s$^{-1}$ for 60 min after passing through all velocities, except that of fatigue. For each velocity, new fish were used, which swam all previous velocities until reaching the target velocity. Different letters indicate significant differences between velocities for the same species ($P < 0.05$). Data are shown as mean ± sem ($n = 6$).

**Table 1 Activity of lactate dehydrogenase (LDH) in the white and red muscle of matrinchã and tambaqui exercised until fatigue.** Unity of measure in mols of pyruvate min$^{-1}$ g$^{-1}$. C refers to control, F exercised until fatigue, WM white muscle, RM red muscle. Data are shown as mean ± sem ($n = 6$). ANOVA main effects (AME) are shown below the respective results, considering a $p$-value of 0.05.

| | | White muscle | | Red muscle | |
|---|---|---|---|---|---|
| | | Tambaqui | Matrinchã | Tambaqui | Matrinchã |
| LDH | C | 125.32 ± 14.86 | 197.5 ± 19.17 | 84.08 ± 5.72 | 213.3 ± 21.00 |
| | F | 121.24 ± 15.24 | 194.8 ± 12.00 | 75.72 ± 6.84 | 203.4 ± 17.18 |
| | | AME: species: $P < 0.001$; exercise: $P = 0.187$; interaction: $P = 0.249$ | | AME: species: $P < 0.001$; exercise: $P = 0.624$; interaction: $P = 0.940$ | |

In contrast, in matrinchã, the NOx levels are higher than resting fish only at 3.75 bl s$^{-1}$ ($p < 0.05$), increasing even more at the fatigue. At 2.5 bl s$^{-1}$, the velocity close to that of fatigue of tambaqui, the levels of NOx are similar between species. After 1 h of recovery, NOx levels fall to control levels in tambaqui and matrinchã.

LDH did not change activity at fatigue in either muscle type in either species (Table 1). However, higher LDH activity in matrinchã, compared to tambaqui, was observed in both muscle types, and in all tested conditions ($p < 0.05$).

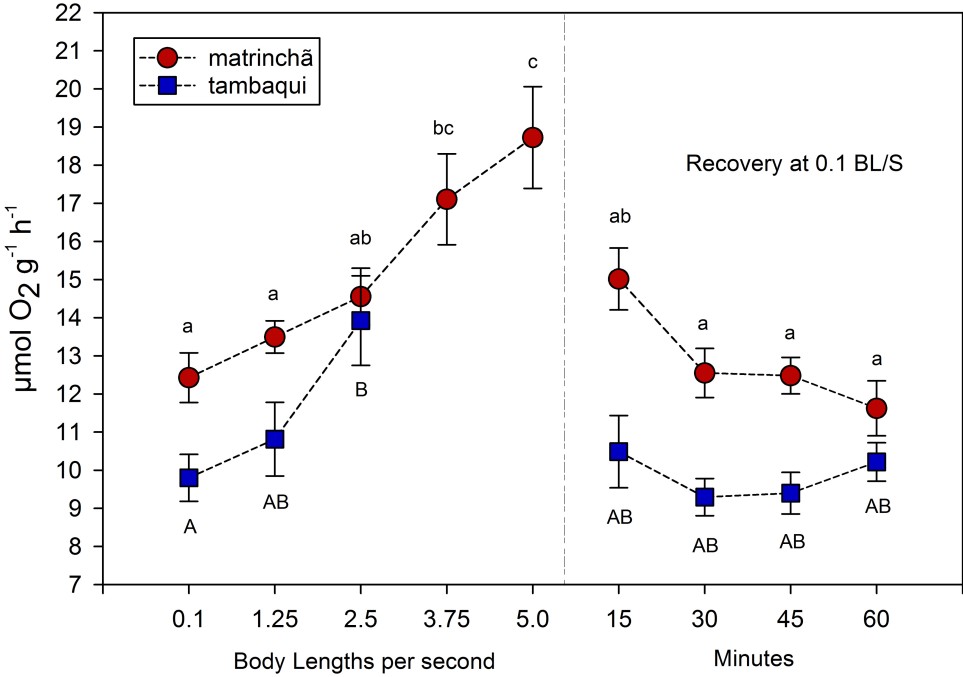

**Figure 2 Oxygen consumption (MO₂) of tambaqui and matrinchã exposed for 30 min to submaximal velocities (body lengths per second; bl s⁻¹).** Recovery refers to a new fish swimming at 0.1 bl s⁻¹ for 15, 30, 45 and 60 min after passing through all velocities, except that of fatigue. For each velocity, new fish were used, which swam all previous velocities until reaching the target velocity. Different letters indicate significant differences between velocities for the same species ($P < 0.05$). Data are shown as mean ± sem ($n = 6$).

### Fish exposed to hypoxia

LDH activity increased in response to hypoxia in the white muscle only in tambaqui, but activities were always higher in matrinchã ($P < 0.05$, Table 2). In contrast, in red muscle, hypoxia caused a decrease in LDH activity in both species ($P < 0.05$); although activities for matrinchã were always higher ($P < 0.05$, Table 2) than those observed for tambaqui.

Lactate levels in white muscle were similar in both species after hypoxia, but only in tambaqui were higher than control ($P < 0.05$, Table 2). In red muscle, hypoxia did not cause an increase in lactate levels, though levels were always higher in matrinchã than in tambaqui ($P < 0.05$, Table 2).

## DISCUSSION

### Does tambaqui accumulate lactate faster than matrinchã during the Ucrit test?

As well as in previous analysis (*Ferreira, 2006*; *Ferreira, Oliveira & Val, 2010*), in the present experiment, the swimming performance (Ucrit) of matrinchã was higher than of the tambaqui. Interestingly, at the first velocity (1.25 bl s⁻¹), the muscle lactate levels were lower than resting fishes (0.1 bl s⁻¹) in both species. *Gladden (2004)* pointed that, in general, lactate is not only produced by intense exercise, but it is also continuously

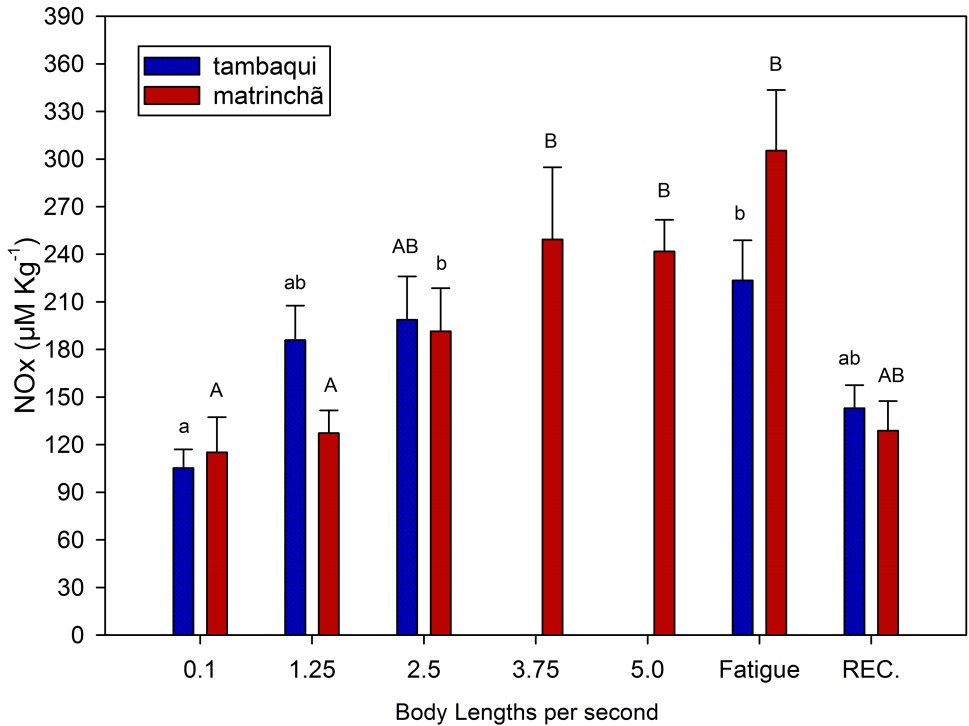

**Figure 3** **NOx (nitrites plus nitrates) in red muscle of tambaqui and matrinchã exposed for 30 min to submaximal velocities (body lengths per second; bl s⁻¹) and at fatigue.** Recovery (REC.) refers to the new fish swimming at 0.1 bl s⁻¹ for 60 minutes after passing through all velocities, except that of fatigue. For each velocity, new fish were used, which swam all previous velocities until reaching the target velocity. Different letters indicate significant differences between velocities for the same species ($P < 0.05$). Data are shown as mean $\pm$ sem ($n = 6$).

**Table 2** **Activity of the enzyme lactate dehydrogenase (LDH) and concentration of lactate in the white and red muscle of matrinchã and tambaqui exposed to hypoxia (0.5 mg/L) and open access to the water surface.** Unity of measure for LDH in mol pyruvate min⁻¹ g⁻¹; for lactate in mmol kg⁻¹ in fresh tissue. LAC refers to lactate, WM white muscle, RM red muscle. Data are shown as mean $\pm$ sem ($n = 6$). Different uppercase letters indicate differences ($P < 0.05$) for treatment within the same species. ANOVA main effects (AME) are shown below respective result, considering a $p$- value of 0.05. Data are shown as mean $\pm$ sem ($n = 6$)

| | | White muscle | | Red muscle | |
|---|---|---|---|---|---|
| | | **Tambaqui** | **Matrinchã** | **Tambaqui** | **Matrinchã** |
| LDH | C | 110.96 $\pm$ 8.12A | 275.4 $\pm$ 12.42 | 102.84 $\pm$ 9.48 | 323.2 $\pm$ 18.56 |
| | H | 149.3 $\pm$ 9.92B | 247.9 $\pm$ 19.27 | 63.76 $\pm$ 5.08 | 270.0 $\pm$ 21.06 |
| | | AME: species: $P < 0.001$; hypoxia: $P = 0.312$; interaction: $P = 0.041$ | | AME: species: $P < 0.001$; hypoxia: $P < 0.001$; interaction: $P = 0.384$ | |
| LAC | C | 7.74 $\pm$ 0.80A | 9.53 $\pm$ 0.84 | 6.15 $\pm$ 0.76 | 11.07 $\pm$ 1.07 |
| | H | 15.06 $\pm$ 1.67B | 12.54 $\pm$ 1.08 | 6.76 $\pm$ 0.91 | 10.10 $\pm$ 1.04 |
| | | AME: species: $P = 0.630$; hypoxia: $P < 0.001$; interaction: $P < 0.001$ | | AME: species: $P < 0.001$; hypoxia: $P = 0.585$; interaction: $P = 0.862$ | |

generated by glycolysis, even under moderate conditions. According to the authors, it does not accumulate because it is continuously consumed by adjacent cells or is oxidized in the circulatory system, or yet reconverted to pyruvate before being stored as glycogen in red muscle. Thus, lactate accumulation occurs when there is an imbalance between production and use (*Milligan & Girard, 1993*; *Gladden, 2004*). Therefore, at the velocity of 1.25 bl s$^{-1}$ a moderate and predominately aerobic exercise would stimulate lactate oxidation, resulting in lower plasma and muscle lactate levels if compared to resting fishes.

At 2.5 bl s$^{-1}$, lactate attained the highest concentration in white muscle of tambaqui (Fig. 1), keeping the same concentration until fatigue has occurred, a few minutes later. On the other side, the matrinchã at the moment of fatigue, in addition to having attained a higher swimming performance (Ucrit), presented 61% higher lactate levels in white muscle when compared to fatigued tambaqui. However, over the three first velocities of the test, when tambaqui was not yet fatigued, both species presented similar lactate levels. That is, tambaqui did not increase lactate levels faster than matrinchã (Fig. 1); actually, lactate increased at similar rates in both species. Therefore, our first hypothesis is proved true, since that is not a faster increase of lactate levels in white muscle over the increments of velocities that account for the early fatigue of tambaqui compared to matrinchã.

### Could resistance to lactate accumulation contribute to the better swimming performance of matrinchã compared to tambaqui?

Despite to importance of aerobic capacity to animal performance, is ultimately the over-usage of anaerobic metabolism and lactate accumulation that causes muscle fatigue (reviewed by *Gladden, 2004*). For example, during the Ucrit test, the maximum swimming speed is always related to the higher concentrations of lactate in muscles of fishes ((*Hammer, 1995*; *Burgetz et al., 1998*; *Richards et al., 2002*; *Peake & Farrell, 2004*). As seen in the previous section, matrinchã do not increase muscle lactate levels faster than matrinchã, but the final levels at the moment of fatigue are 61% higher in matrinchã, in addition to having a Ucrit 74% higher. Anyway, it is still not known all the traits that could account for this higher resistance of matrinchã to lactate accumulation, which in turn could support a better swimming performance.

It would be plausible to think that a higher metabolic rate would allow fishes to use produced lactate as an aerobic substrate (*Brookes, 1986*; *Gladden, 2004*; *Omlin, Langevin & Weber, 2014*), improving swimming performance. This capacity of using lactate as aerobic fuel at high velocities could not be as good in tambaqui as in matrinchã, probably due to the lower aerobic scope (the difference between the maximum and the minimum metabolic rate; *Clark, Sandblom & Jutfelt, 2013*). Comparing the aerobic scope of both species, it is 31% higher in matrinchã than tambaqui (4.55 and 6.12 $\mu$mol g$^{-1}$ h$^{-1}$ respectively).

During moderate to high-intensity exercise, glycolytic muscle fibers produce lactate, and some of them escape and concentrate in circulation, being oxidized by neighboring muscle fibers. After being taken up, the lactate oxidation rate depends on the metabolic rate of both, exercising and resting muscles (*Gladden, 2004*).

*Omlin, Langevin & Weber (2014)*, studying rainbow trout, clarify the importance of aerobic metabolism in lactate elimination in exercised muscle and shows that oxidation is

responsible for most of the increase in lactate removal observed during intense swimming. *Brookes (1986)* first pointed the possibility of a "lactate shuttle", and explains that during human rest, approximately 50% of lactate disposal take place through oxidation, whereas during strenuous exercise (around 50–75% $VO_2$ max) the active cells use about 75–80% of lactate. The high usage rate of lactate during exercise in humans indicates that it is an important alternative energy source during increased exercise. *Omlin, Langevin & Weber (2014)*, studying trout, observed that graded up exercise to critical swimming speed (Ucrit) causes the stimulation of lactate appearance (67%) as well as lactate clearance (41%). The authors suggest that lactate accumulates because it cannot be processed as rapidly by oxidative tissues (red muscle, heart, gills, and brain) as it is produced by anaerobic glycolysis in white muscle.

However, classical and recent works have shown that the relationship between the metabolic rate and swimming speed is exponential at high velocities (*Brett, 1964*; *Webb, 1997*; *Di Santo, Kenaley & Lauder, 2017*); that is, as the speed of water increases, more and more oxygen is consumed per unit of time. Thus, based on this type of relationship, it would be expected that the slightly larger aerobic scope of the matrinchã would be related to an even smaller advantage in Ucrit. However, the opposite occurs, and a small advantage in the aerobic scope of matrinchã (31% higher—4.55 against 6.12 $\mu$mol g$^{-1}$ h$^{-1}$) is related to a quite significant advantage in Ucrit (74% higher—3.5 against 6.0 bl s$^{-1}$). These results suggest that other traits of the matrinchã, apart from the aerobic scope and the supposed higher capacity of using lactate as metabolic fuel, also support the higher Ucrit in this species.

In this experiment, NOx (as an indicator of nitric oxide) was measured to indicate the improvement of the circulatory system in both species and its possible contribution to postpone lactate production and fatigue. Swimming capacity of fishes has been related to the physiological mechanisms causing blood flow increases (reviewed by *Webb, 1997*). Therefore, it is possible that just like in mammals, nitric oxide could improve $O_2$ delivery and $CO_2$ removal from muscles, what could also improve lactate oxidation. However, this subject has never been addressed in fishes. In the present work, just like metabolic rate and lactate levels, NOx presented similar levels (Fig. 3) in the two analyzed species at 2.5 bl s$^{-1}$ (the velocity closes to tambaqui's fatigue), suggesting that the nitric oxide does not have any significative contribution in postponing lactate production and fatigue in matrinchã. However, Nitric Oxide could be an aid for the better swimming performance of fish in general as it acts improving the cardio-respiratory system. More studies are necessary to understand the relationship between nitric oxide in muscles and swimming performance in fish.

Considering that matrinchã is more fusiform, while tambaqui is a more rounded fish, the body shape and hydrodynamics could also be an aid to swimming performance by postponing lactate accumulation and fatigue. In tambaqui, it seems that there is a subtle increase in metabolic rate from 1.25 to 2.5 bl s$^{-1}$, just before fatigue, while in matrinchã the increases were more gradual over the entire test. That is, while in matrinchã the lactate levels increased 29% from 1.25 to 2.5 bl s$^{-1}$, in tambaqui, it increased by 102% in the same period. This subtle increase suggests poor hydrodynamics of tambaqui at 2.5 bl s$^{-1}$; however, the data are not enough to allow us to make any strong conclusion

about it. Matrinchã is a rheophilic fish, living in flowing water bodies (*Zaniboni-Filho, Reynalte-Tataje & Weingartner, 2006*), while tambaqui is commonly found in slow water. The preferred habitat could help to explain the evolutionary forces that shaped the body and differences in swimming performance in these species, a matter for future research. Anyway, eventual effects of hydrodynamic or other traits, as glycogen content and its usage should be tested by specific experiments, not addressed in this study.

Apart from the other traits and physiological improvements that could support matrinchã to attain a higher Ucrit, they do not conflict or invalidate our hypothesis. Our results showed that both species presented similar lactate levels at the time of tambaqui's fatigue and that matrinchã attained much higher lactate levels in muscle at the end of the test. Therefore, our second hypothesis is proved false, and we cannot discard the possibility that differences in swimming performance could be supported by resistance to lactate accumulation in matrinchã.

## Are the anaerobic metabolic adjustments different between these species?

To better understand the anaerobic capacity of each muscle type, and the capacity of reconversion of lactate to pyruvate, lactate dehydrogenase (LDH) activity was measured in red and white muscles in fatigued fishes. As seen in Table 1, LDH activity is higher in matrinchã in all analyzed conditions. The enzyme lactate dehydrogenase is responsible for allowing the generation of ATP in the absence of oxygen (fermentation) since that flux through glycolysis can continue in the absence of NADH oxidation by the electron transport chain (ETC) inside the cell mitochondria. However, LDH can reconvert lactate to pyruvate, when oxygen is available again (*Hochachka & Somero, 1984*; *Wieser et al., 1987*; *Almeida-Val et al., 2011*).

The higher LDH activity in red muscles could be useful to deal with the higher lactate levels after the exercise, as observed for matrinchã. Since lactate accumulated in white muscles (Fig. 1) with no changes of LDH activity in this tissue at the fatigue in both species, the accumulation of lactate seems to have occurred mainly due to the difficulty of aerobic tissues to oxidize it, and not due to additional production from the white muscles. Also, it is possible that at low velocities, white muscle has a low contribution to power generation thus increasing LDH activity is not required.

In the hypoxia exposure test, the experimental protocol was conducted to fastly decrease the oxygen levels inside the chamber (about 30 min). We have used this procedure to make the fishes to face similar conditions that occur during intense swimming at the Ucrit test, in that it has to start quickly using the anaerobic metabolism, with just a few minutes to adapt to this new condition.

Higher lactate levels were observed in control group of matrinchã (normoxic water), in the hypoxia experiment, if compared to resting levels from incremental velocity experiment. This higher level of lactate, even in normoxic conditions, could be caused by the constraint of the animals in two-liter experimental tanks. Since this fish has a higher metabolic rate, the chamber constraint could be a stressful factor resulting in higher lactate levels. As a result, any possible effects of hypoxia on the lactate in the white muscle of matrinchã could

have been masked. Despite this, the hypoxia experiment gave us some interesting insights about the anaerobic metabolism based on LDH activity, which could not be ignored.

Despite high lactate levels in white muscle of both species after hypoxia, LDH activity does not change for matrinchã, contrasting with increased LDH activity in tambaqui. Nevertheless, LDH activity is greater in matrinchã in every experimental condition, what could indicate a high capacity for anaerobic power generation. However, having a higher basal activity in LDH activity may mean that there is no more capacity to increase further LDH activity during hypoxia, as seen in the present experiment. On the other hand, LDH decreased in the red muscle of both species after hypoxia, (Table 2) thought it was always higher in matrinchã. Since red muscle is predominantly aerobic, higher activity of LDH indicates a higher capacity to convert lactate back to pyruvate. That is, in addition to being able to oxidize lactate coming from white muscle, the red muscle could be a site for the clearance of lactate by reconversion to pyruvate.

LDH presented different trends in muscles of tambaqui under hypoxia: in white muscle, LDH activity seems to increase to generate energy anaerobically from pyruvate; while in red muscle, it decreased, since that without oxygen, it is not useful to produce energy or to convert lactate back to pyruvate (Table 2). A similar trend was also observed in the red muscle of matrinchã, that is, since that red muscle is predominantly aerobic, it does not make sense to increase LDH activity under hypoxia (Table 2). For example, in the heart of fishes and many other animals, the isoform LDH-B predominates, which is inhibited by the accumulation of its substrate, the pyruvate. The LDH-B isoform would protect such sensitive tissues against the aggressive effects of lactate accumulation under hypoxia (*Gutfreund et al., 1968*; *Almeida-Val & Val, 1993*). The reduction in LDH activity, under hypoxia, could be reached by allosteric effectors (*Brown & Christian, 1974*), by shifting between dimer and tetramer forms (*Yamamoto & Storey, 1988*), or by peptides that affect the interaction between LDH subunits (*Döbeli & Schoenenberger, 1983*).

Since the activity of LDH increases in the white muscle of tambaqui due to exposure to hypoxia (Table 2) but does not increase during intense exercise (Table 1), it is possible that two different pathways cause the accumulation of lactate in these conditions. Under intense exercise, lactate may accumulate mainly due to reduced oxidation; while under hypoxia, a sharp increase in lactate production may have occurred, as result of the intense use of anaerobic metabolism for energy production.

The results obtained from exercise and hypoxia experiments motivate the rejection of the third hypothesis, i.e., the metabolic rate and the anaerobic metabolism are not similar between the species at rest and fatigue. In fact, there are some metabolic differences that, in addition to lactate resistance, could support the better swimming performance of matrinchã. Conceivably, all these factors are interconnected, that is, the higher metabolic rate and anaerobic capacity of matrinchã could favor the accumulation and oxidation of muscle lactate, avoiding fatigue.

## CONCLUSIONS

The results suggest that matrinchã is more resistant than tambaqui regarding lactate accumulation during the Ucrit test, and it is possible that the higher metabolic rate

could ensure a better usage of lactate as aerobic fuel, postponing fatigue. The anaerobic metabolism, as well as the other parameters, indicates a possible tradeoff between being a good swimmer or hypoxia-resistant fish, for matrinchã and tambaqui, respectively. More studies are needed to clarify why matrinchã is a better swimmer than tambaqui, especially on the ability of these species to oxidize lactate, the effects of exercise on muscle acidification, and on the hydrodynamics.

## ACKNOWLEDGEMENTS

We thank Maria de Nazaré Paula da Silva for assisting us in the laboratory analyses.

### Funding

This work was funded by a joint grant from the Brazilian National Research Council (CNPq) and the Amazonas State Research Foundation (FAPEAM) to INCT-ADAPTA. Adalberto L. Val is the recipient of a research fellowship from the Brazilian National Research Council (CNPq). Marcio S. Ferreira is the recipient of a Post-Doctoral fellowship from the Brazilian Centre for Improvement of Higher Education Personnel (CAPES). The funders had no role in study design, data collection and analysis, decision to publish, or preparation of the manuscript.

### Grant Disclosures

The following grant information was disclosed by the authors:
Brazilian National Research Council (CNPq).
Amazonas State Research Foundation (FAPEAM).
Brazilian Centre for Improvement of Higher Education Personnel (CAPES).

### Competing Interests

The authors declare there are no competing interests.

### Author Contributions

- Marcio S. Ferreira conceived and designed the experiments, performed the experiments, analyzed the data, contributed reagents/materials/analysis tools, prepared figures and/or tables, authored or reviewed drafts of the paper, approved the final draft.
- Paulo H.R. Aride analyzed the data, contributed reagents/materials/analysis tools, approved the final draft.
- Adalberto L. Val conceived and designed the experiments, analyzed the data, contributed reagents/materials/analysis tools, authored or reviewed drafts of the paper, approved the final draft.

### Animal Ethics

The following information was supplied relating to ethical approvals (i.e., approving body and any reference numbers):

Experimental work was approved by the Ethics Committee on Animal Experiments of INPA (protocol number: 020/2013) and conformed to national animal care regulations.

## Data Availability

The raw data are provided in Data S1.

## Supplemental Information

Supplemental information for this article can be found online at http://dx.doi.org/10.7717/peerj.5719#supplemental-information.

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
