# Peer review of "Could resistance to lactate accumulation contribute to the better swimming performance of Brycon amazonicus when compared to Colossoma macropomum?"

_PeerJ, doi:10.7717/peerj.5719_

## Round 0.1 · original submission · Major Revisions

I have now received comments from three reviewers. As you can see, two recommended MINOR REVISIONS and one recommended REJECT. When such situation arises, the Editor find himself in a very uncomfortable position. By reading the reasons of rejection, I call attention for the following lines "Although the experiment design is appropriate for the aim of the study, the general hypothesis of the paper is fundamentally flawed and the conclusions are not supported by the data collected." These lines are telling me that your experiment is valid, but the reviewer does not agree with your hypothesis. Based on that, I opted to give you a chance to respond to the reviewer explaining why you disagree with him/her or by reformulating your paper according the reviewer's suggestion. The way you answer to him/her will be crucial to the acceptance or rejection of your paper. Good luck with your revision!

Reviewer 1 ·

Basic reporting

Figures are clear and appropriate. English grammar requires work throughout. The text is generally clear but it requires substantial copy editing to meet the language requirements of the journal.

Experimental design

Although the experiment design is appropriate for the aim of the study, the general hypothesis of the paper is fundamentally flawed and the conclusions are not supported by the data collected. The authors argue that tolerance to white muscle lactate accumulation underlies the better swimming performance of matrincha compared to tambaqui but a substantial literature-base going back decades has shown that aerobic capacity, supported by cardiorespiratory function, is likely the most critical factor in determining athletic performance in fish (and most other vertebrates). The authors themselves discuss why a single measurement of lactate concentration in white muscle cannot be used to predict anaerobic capacity, since the simultaneous oxidation of lactate can also be upregulated.

The data collected are valid and the experiments appear to have been carried out rigorously and analyzed appropriately. The hypoxia experiment was not relevant to the study.

Validity of the findings

The data collected in the study provide a clear, straightforward answer to the question of why matrincha is a better swimmer: Figure 2 shows that matrincha has an aerobic scope (the difference between minimum and maximum MO2) almost twice that of tambaqui. This illustrates that matrincha has a much greater cardiorespiratory capacity, likely a much greater capacity for oxidative phosphorylation, a much greater capacity to deliver O2 to working muscles and therefore avoid anaerobic metabolism, and finally a much greater capacity to oxidize any lactate produced in that working muscle. Unfortunately, the remainder of the data presented in the paper is not particularly helpful in answering the question posed.

The authors attempt to draw conclusions about plasticity of anaerobic capacity during swimming by measuring the Vmax of LDH, and not the actual flux of pyruvate to lactate. Substantial upregulation of maximum LDH activity requires the synthesis of new LDH protein, which is unlikely to occur during the energy limited conditions associated with a maximum swim performance test.

Additional comments

I would encourage the authors to re-frame the paper in a different way, perhaps focusing on the differences aerobic scope (recognizing that the maximum MO2 measurements may be slightly underestimated due to the logistics of the setup).

Line 50 – specify aerobic metabolic rate, since anaerobic metabolism will be occurring but you are only measuring O2 consumption.
Line 95-97 - clarify - does this mean that their swimming performance declined after exposure to hypoxia or that they were poor endurance swimmers in general?

Line 103-104 - Does environmental hypoxia increase white muscle [lactate] in these animals? Clarifying this point would help the reader to better understand the rationale for the current study. If hypoxia only increases plasma lactate, then that can act as an efficient fuel for swimming during normoxic exercise.

Line 115-117 - Lactate in white muscle is probably largely produced from on-board glycogen stores. Is swimming limited by glycogen content in these species?

Line 126-128 - but lactate is largely retained in the white muscle in fish, since they lack a Cori cycle. If lactate was released from the white muscle, it would be hugely concentrated in the blood since the white muscle makes up such a large proportion of total body mass in fish.

Line 136-138 - not increased cardiorespiratory capaciy - just cardiorespiratory capacity. Adding the 'increased' there implies that maximum cardiorespiratory function can be instantly increased by exercise.

Line 151-153 - This hypothesis seems to conflict with the background given that tambaqui will fatigue more quickly than matrincha.

Line 160-162 - phrasing is not accurate here. Contribution of NOx to facilitating cardiac scope would be more accurate.

Line 229-238 - There is no swimming post hypoxia test. The intro of the paper refers to that test often and makes it seem like that will be tested here.

In results text – if it is stated that something is ‘different’, make sure the statement is backed up by statistical testing and a p-value.

Line 270-272 -The MO2 data pretty clearly show that matrincha has a much greater aerobic scope than tambaqui, even though the maximum MO2 was not necessarily measured for either species. This alone largely explains why matrincha is a better swimmer.

Line 286-293 - The hypoxia experiment really doesn't add to this study. It is not directly relevant to the main question being asked and the data is not useful.

Line 300-302 - this implies that you confirmed the ability to accumulate lactate explains the better swimming ability in matrincha, which was not proven.

Line 308-310 – lactate cannot be both oxidized and stored - it is one or the other.

Line 322-328 - hydrodynamics were not addressed in this study so this conclusion is inappropriate.

Line 333-334 - It is confirmed by the MO2 data - swimming velocities are secondary. This interpretation is problematic in that it makes the assumption that higher metabolic rate allows higher swimming velocities but the entire premise of the paper was to test if lactate tolerance allowed higher swimming velocities.

Line 340- 345 – No, this is incorrect. It just has a lower maximum MO2 (lower aerobic scope). It is not as athletic. The metabolic rate data absolutely does explain the differences in swimming capacity. Tambaqui likely has both lower aerobic and anaerobic capacity (at least in the white muscle, which is not critical for hypoxia tolerance).

Line 347-359 - most of these parameters were not measured so it is not appropriate to spend this much time discussing them.

Line 365-367 - but NOx did continue to increase (although not statistically significantly) in the matrincha. It could contribute.

Line 375-376 - no, it most certainly is not. The LDH reaction does not produce ATP, it oxidizes NADH to NAD+ so that flux through glycolysis can continue in the absence of NADH oxidation by the ETC.

Reviewer 2 ·

Basic reporting

No comment.

Experimental design

Regarding the experiment with fish exposure to the hypoxic condition, it is important that the authors justify the reason for having lowered the oxygen concentration of the chamber so quickly (in 30 min). Also justify why the fish had been exposed to hypoxia for only 1 hour.

Validity of the findings

No comment.

Reviewer 3 ·

Basic reporting

Some references listed at the end of the manuscript do not follow the journal standard. I suggest the authors review more carefully.

Experimental design

The experimental design is well described and detailed.

Validity of the findings

No comments

Additional comments

The manuscript “Does lactate or metabolic rate account for the better swimming performance of Brycon amazonicus when compared to Colossoma macropomum” investigating if matrinchã can keep swimming even with high lactate levels in muscle and analyzed the differences in aerobic and anaerobic metabolism between the fish species. The manuscript is constructed properly. The experiment is planned correctly, based on proper methods and does not arise any ethical or content-related reservations. Most of the results are presented carefully in very clear form. The presentation of the results in tables and figures can be improved. In this manuscript is required minor revision before its acceptance.
Comments
In the abstract, the name of the species must be separated.
I suggest putting keywords at the end of the abstract.
P5, L 4: Use superscript in identifying authors
P10, L 177: Change mg L-1 by mgL-1
P11, L 210: Describe the compound name MS-222
P12, L 235: Change ASR (aquatic surface respiration) by aquatic surface respiration (ASR)
The authors use many old references and many could be replaced.
P21, L513; P22, L537: The name of the journal is not in the rules.
P25: Describe in Figure 1 which means the x-axis

---

## Round 0.2 · Major Revisions

I am asking again for Major Revisions simply because the following comment of one of the reviewers was not addressed directly in the rebuttal letter, and this is a fundamental step towards the acceptance of this manuscript:

"Although the experiment design is appropriate for the aim of the study, the general hypothesis of the paper is fundamentally flawed and the conclusions are not supported by the data collected. The authors argue that tolerance to white muscle lactate accumulation underlies the better swimming performance of matrincha compared to tambaqui but a substantial literature-base going back decades has shown that aerobic capacity, supported by cardiorespiratory function, is likely the most critical factor in determining athletic performance in fish (and most other vertebrates). The authors themselves discuss why a single measurement of lactate concentration in white muscle cannot be used to predict anaerobic capacity, since the simultaneous oxidation of lactate can also be upregulated".

---

## Round 0.3 · accepted · Accept

The authors have satisfactorily addressed the comments from the three reviewers.

The new title reflects the paper contents and the conclusions are substantiated by the results. The discussion is greatly improved and it now adequately considers other factors that are known to affect swimming performance in fish, namely the aerobic scope.

#